# ISOLATING SOURCES OF DISENTANGLEMENT IN VARIATIONAL AUTOENCODERS

**Tian Qi Chen, Xuechen Li, Roger Grosse & David Duvenaud**
University of Toronto
Vector Institute
Toronto, Canada
{`rtqichen,lxuechen,rgrosse,duvenaud`}`@cs.toronto.edu`

## ABSTRACT

We decompose the evidence lower bound (ELBO) to show the existence of a total correlation term between latents. This motivates our $\beta$-TCVAE (Total Correlation Variational Autoencoder), a refinement of the $\beta$-VAE for learning disentangled representations without supervision. The $\beta$-TCVAE is equivalent to the recently proposed FactorVAE in objective but optimized differently. We further propose a principled classifier-free measure of disentanglement called the Mutual Information Gap (MIG). We show a strong relationship between total correlation and disentanglement.

## 1 INTRODUCTION

Many state-of-the-art methods for learning disentangled representations without supervision are based on increasing the penalty on parts of an existing objective. Higgins et al. (2017) argue that encouraging independence in latents induces disentanglement. However, more theoretical evidence is required to show factorial representations lead to disentanglement.

In this paper, we decompose the evidence lower bound (ELBO) into terms describing: 1) the independence of latents and 2) the mutual information between latents and the observed. We analyze the $\beta$-VAE objective and propose the improved $\beta$-TCVAE, which is equivalent to FactorVAE (Kim & Mnih, 2017) in objective but optimized differently. To evaluate our method, we propose the Mutual Information Gap (MIG), a principled information-theoretic quantity that measures disentanglement. Compared to the $\beta$-VAE, we discover that the $\beta$-TCVAE is more reliable, can reveal more features, and exhibit a stronger correlation between factorial and disentangled representations.

## 2 SOURCES OF DISENTANGLEMENT IN THE ELBO

We introduce the $\beta$-TCVAE with a refined decomposition showing that an additional term describing the independence between the latents appears in the ELBO when a factorial prior is used. This decomposition can be seen as a special case of disentanglement under the information-bottleneck framework (Achille & Soatto, 2018).

### 2.1 ELBO TC-DECOMPOSITION

Following Hoffman & Johnson (2016), we identify each training example with a unique integer index and define a uniform random variable on $\{1, 2, ..., N\}$. We define $q(z|n) = q(z|x_n)$ and $q(z, n) = q(z|n)p(n) = q(z|n)/N$. We refer to $q(z) = \sum_{n=1}^{N} q(z|n)p(n)$ as the *aggregated posterior* following Makhzani et al. (2016). We decompose the mean-of-KL in the ELBO assuming a factorized prior (see Appendix 5.2 for detailed derivation):

$$\mathbb{E}_{p(n)}\Big[D_{\text{KL}}\big(q(z|n)||p(z)\big)\Big] = \underbrace{D_{\text{KL}}\left(q(z, n)||q(z)p(n)\right)}_{\text{(i) Index-Code MI}} + \underbrace{D_{\text{KL}}\big(q(z)||\prod_j q(z_j)\big)}_{\text{(ii) Total Correlation}} + \underbrace{\sum_j D_{\text{KL}}\left(q(z_j)||p(z_j)\right)}_{\text{(iii) Dimension-wise KL}}$$

$$(1)$$

We hypothesize that a low total correlation is the main reason $\beta$-VAE achieves empirical success in learning disentangled representations. However, we emphasize the $\beta$-VAE also encourages the model to discard information from the latents since it penalizes the index-code mutual information.

## 2.2 $\beta$-TCVAE

It is possible to assign different penalty weights to terms in (1), but instead, we penalize only the total correlation term. The $\beta$-TCVAE objective is:

$$\mathcal{L}_{\beta\text{-TC}} := \frac{1}{N} \sum_{n=1}^{N} \Big( \mathbb{E}_{q(z|n)}[\log p(n|z)] \Big) - I(z; n) - \boldsymbol{\beta} \, \mathrm{D}_{\mathrm{KL}}\big(q(z) || \prod_j q(z_j)\big) - \sum_j \mathrm{D}_{\mathrm{KL}}\left(q(z_j) || p(z_j)\right) \quad (2)$$

where $\beta > 1$. This objective is the same as proposed by Kim & Mnih (2017) with $\gamma = \beta - 1$. However, the way we optimize the objective is different. The objective allows us to train latent variable models that encourage an independent aggregated posterior, which we argue increases the chances of obtaining a disentangled representation. Note that different penalization coefficients can be used for all terms in the decomposition, but we keep the other coefficients at one to ensure a valid lower bound on the likelihood.

The decomposed expression (1) requires the evaluation of $q(z) = \mathbb{E}_{p(n)}[q(z|n)]$, which depends on the entire empirical dataset. As such, it is undesirable to compute it exactly during training. We discuss one possible way to estimate this stochastically using a minibatch of samples without any added hyperparameters. We propose using a minibatch importance sampled version for estimating the function $\log q(z)$ during training.[1] (See Appendix section 5.3 for a detailed description).

$$\mathbb{E}_{q(z)}[\log q(z)] \approx \frac{1}{M} \sum_{i=1}^{M} \left[ \log \frac{1}{NM} \sum_{j=1}^{M} q(z(n_i)|n_j) \right] \quad (3)$$

## 3 MEASURING DISENTANGLEMENT WITH MIG

We propose a metric based on the empirical mutual information between latents and ground truth factors. The full metric we call *mutual information gap* (MIG) is then

$$\frac{1}{K} \sum_{k=1}^{K} \frac{1}{H(v_k)} \left( I_n\big(z_{j^{(k)}}; v_k\big) - \max_{j \neq j^{(k)}} I_n(z_j; v_k) \right) \quad (4)$$

where $j^{(k)} = \mathrm{argmax}_j \, I_n(z_j; v_k)$ and $K$ is the number of known factors. MIG is between 0 and 1.

The gap in our formulation (4) defends against two important cases. The first case is related to rotation of the factors, or more precisely axis-alignment. The second case is related to compactness of the representation. If one latent variable reliably models a ground truth factor, then it is unnecessary for other latent variables to also be informative about this factor.

## 4 EXPERIMENTS

### 4.1 FACTORIAL VS. DISENTANGLED REPRESENTATIONS

To understand the robustness of VAE, $\beta$-VAE, InfoGAN Chen et al. (2016), and $\beta$-TCVAE in learning disentangled representations, we show box plots depicting the quartiles of the MIG score distribution for various methods.

Figure 1 shows the total correlation and the MIG disentanglement metric for varying values of $\beta$ trained on the dSprites and faces datasets averaged over 40 random initializations. For models trained with $\beta$-TCVAE, the correlation between average TC and average MIG is $-0.95$, while models trained with $\beta$-VAE have a lower correlation. For the dSprites data, we see that with higher values of $\beta$, $\beta$-VAE seems to stop reducing the TC among latents while disentanglement further

---

[1]The same argument holds for the term $\prod_j q(z_j)$ and a similar estimator can be constructed.

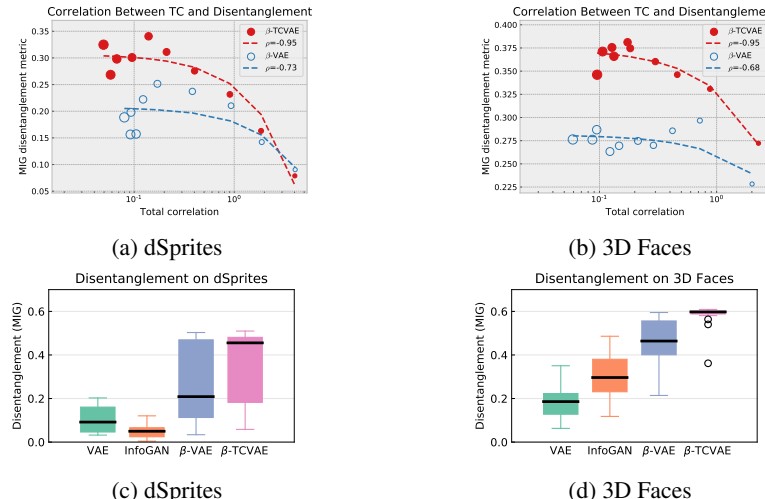

(a) dSprites          (b) 3D Faces

(c) dSprites          (d) 3D Faces

Figure 1: (a) and (b) are scatter plots of the average MIG and TC per value of $\beta$. Larger circles indicate a higher $\beta$. On average, $\beta$-TCVAE makes better use of low total correlation scores to reach higher disentanglement. (c) and (d) are distributions of disentanglement score (MIG).

decreases. Meanwhile, $\beta$-VAE achieves a lower total correlation than $\beta$-TCVAE on the 3D faces dataset, possibly an indication of latents regressing towards the prior and becoming inactive. This is also strong evidence for our hypothesis of $\beta$-VAE that large values of $\beta$ reduces the index-code mutual information in the latents, leading to a worse generative model.

## 4.2 QUALITATIVE RESULTS

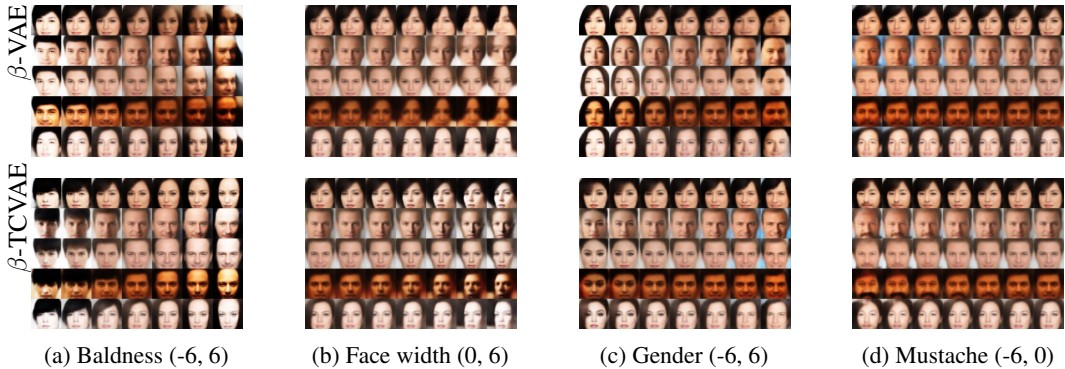

(a) Baldness (-6, 6)     (b) Face width (0, 6)     (c) Gender (-6, 6)     (d) Mustache (-6, 0)

Figure 2: Qualitative comparisons on celebA. Traversal ranges are shown in parentheses. Some attributes are only manifested in one direction of a latent variable, so we show a one-side traversal.

**CelebA**   Figure 2 shows 4 out of more than 10 attributes that are discovered by the $\beta$-TCVAE without supervision (see more in Appendix). We traverse in a large range to show the effect of generalizing the represented semantics of each variable. The representation learned by $\beta$-VAE is entangled with nuances, which can be shown when generalizing to low probability regions. For instance, it has difficulty rendering complete baldness or narrow face width, whereas the $\beta$-TCVAE shows meaningful extrapolation. The extrapolation of the gender attribute of $\beta$-TCVAE shows that it focuses more on gender-specific facial features, whereas the $\beta$-VAE is entangled with many irrelevances such as face width. The ability to generalize beyond the first few standard deviations of the prior mean implies that the $\beta$-TCVAE model can generate rare samples such as *bald or mustached females*

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

## 5 APPENDIX

### 5.1 RELATED WORK

We focus on discussing the learning of disentangled representations in an unsupervised manner. Nevertheless, we note that inverting generative processes with known disentangled factors through weak supervision has been pursued by many, where the goal in this case is not perfect inversion but to distill into a simpler representation Hinton et al. (2011); Kulkarni et al. (2015); Cheung et al. (2014); Karaletsos et al. (2016); Vedantam et al. (2018). Although not explicitly the main motivation, many unsupervised generative modeling frameworks have explored the disentanglement of their learned representations Kingma & Welling (2013); Makhzani et al. (2016); Radford et al. (2015). Prior to $\beta$-VAE Higgins et al. (2017), some have shown successful disentanglement in limited settings with few factors of variation Schmidhuber (1992); Tang et al. (2013); Desjardins et al. (2012); Glorot et al. (2011).

With recent progress in generative modeling, Chen et al. (2016) propose the InfoGAN by modifying the generative adversarial networks objective Goodfellow et al. (2014) to encourage high mutual information between a small subset of latent variables and the observed data. This motivates the removal of the index-code MI term in (1) which is explored in some recent works Makhzani et al. (2016); Kumar et al. (2017); Zhao et al. (2017). However, investigations into generative modeling also claim that a penalized mutual information through the information bottleneck encourages compact and disentangled representations Achille & Soatto (2017); Burgess et al. (2017).

As a means to describe the properties of disentangled representations, factorial representations have been motivated by many Schmidhuber (1992); Ridgeway (2016); Achille & Soatto (2018); Ver Steeg & Galstyan (2015), where sometimes total correlation is even equated as a measurement of disentanglement. In a similar vein, non-linear independent component analysis Comon (1994); Hyvärinen & Pajunen (1999); Jutten & Karhunen (2003) studies the problem of inverting a generative process assuming independent latent factors. Instead of a perfect inversion, we only aim for maximizing the mutual information between our learned representation and the ground truth factors. Simple priors can further encourage interpretability by means of warping complex factors into simpler manifolds. Very recently, Kim & Mnih (2017) have also explored penalizing the total correlation in a variational autoencoder; however, we explicitly acknowledge the existence of this term already in the ELBO assuming a factorial prior. Furthermore, our minibatch importance sampling allows adding arbitrary weights to each term in the decomposition (1) and does not require any additional hyperparameters, whereas the learning algorithm by Kim & Mnih (2017) requires an auxiliary discriminator network to estimate the total correlation term. To the best of our knowledge, we are the first to show a strong relation between factorial representations and disentanglement (see Section 4.1).

## 5.2 ELBO TC-DECOMPOSITION

Proof of the decomposition in (1):

$$
\frac{1}{N} \sum_{n=1}^{N} D_{KL}\big(q(z|x_n)||p(z)\big) = \mathbb{E}_{p(n)}\Big[D_{KL}\big(q(z|n)||p(z)\big)\Big]
$$

$$
= \mathbb{E}_{p(n)}\Big[\mathbb{E}_{q(z|n)}\big[\log q(z|n) - \log p(z) + \log q(z) - \log q(z) + \log \prod_j q(z_j) - \log \prod_j q(z_j)]\big]\Big]
$$

$$
= \mathbb{E}_{q(z,n)}\Big[\log \frac{q(z|n)}{q(z)}\Big] + \mathbb{E}_{q(z)}\Big[\log \frac{q(z)}{\prod_j q(z_j)}\Big] + \mathbb{E}_{q(z)}\Big[\sum_j \log \frac{q(z_j)}{p(z_j)}\Big]
$$

$$
= \mathbb{E}_{q(z,n)}\Big[\log \frac{q(z|n)p(n)}{q(z)p(n)}\Big] + \mathbb{E}_{q(z)}\Big[\log \frac{q(z)}{\prod_j q(z_j)}\Big] + \sum_j \mathbb{E}_{q(z)}\Big[\log \frac{q(z_j)}{p(z_j)}\Big]
$$

$$
= \mathbb{E}_{q(z,n)}\Big[\log \frac{q(z|n)p(n)}{q(z)p(n)}\Big] + \mathbb{E}_{q(z)}\Big[\log \frac{q(z)}{\prod_j q(z_j)}\Big] + \sum_j \mathbb{E}_{q(z_j)q(z_{\setminus j}|z_j)}\Big[\log \frac{q(z_j)}{p(z_j)}\Big]
$$

$$
= \mathbb{E}_{q(z,n)}\Big[\log \frac{q(z|n)p(n)}{q(z)p(n)}\Big] + \mathbb{E}_{q(z)}\Big[\log \frac{q(z)}{\prod_j q(z_j)}\Big] + \sum_j \mathbb{E}_{q(z_j)}\Big[\log \frac{q(z_j)}{p(z_j)}\Big]
$$

$$
= \underbrace{D_{KL}\big(q(z,n)||q(z)p(n)\big)}_{\text{(i) Index-Code MI}} + \underbrace{D_{KL}\big(q(z)||\prod_j q(z_j)\big)}_{\text{(ii) Total Correlation}} + \underbrace{\sum_j D_{KL}\big(q(z_j)||p(z_j)\big)}_{\text{(iii) Dimension-wise KL}}
$$

## 5.3 TRAINING WITH MINIBATCH IMPORTANCE SAMPLING

The decomposed expression (1) requires the evaluation of $q(z) = \mathbb{E}_{p(n)}[q(z|n)]$, which depends on the entire empirical dataset. As such, it is undesirable to compute it exactly during training. We discuss one possible way to estimate this stochastically using a minibatch of samples without any added hyperparameters.[2]

First notice that a naïve Monte Carlo approximation based on a minibatch of samples from $p(n)$ is very likely to underestimate $q(z)$. This can be intuitively seen by viewing $q(z)$ as a mixture distribution where the data index $n$ indicates the mixture component. With a randomly sampled component, $q(z|n)$ is close to 0, whereas $q(z|n)$ would be large if $n$ is the component that $z$ came from. So it is much better to sample this component and weight the probability appropriately.

To this end, we propose using a minibatch importance sampled version for estimating the function $\log q(z)$ during training. When provided with a minibatch of samples $\{n_1, ..., n_M\}$, we can use the estimator

$$
\mathbb{E}_{q(z)}[\log q(z)] \approx \frac{1}{M} \sum_{i=1}^{M} \left[\log \frac{1}{NM} \sum_{j=1}^{M} q(z(n_i)|n_j)\right] \tag{5}
$$

where $z(n_i)$ is a sample from $q(z|n_i)$.

Note that while this minibatch estimator is consistent as $M \to \infty$, it is biased since its expectation is a lower bound[3]. Despite this bias, we note that minibatch importance sampling is crucial for enabling training, and may be advantageous due to the absence of any additional hyperparameters compared to methods that require training auxiliary neural nets Kim & Mnih (2017); Zhao et al. (2017).

First, let $\mathcal{B}_M = \{n_1, ..., n_M\}$ be a minibatch of $M$ indices where each element is sampled i.i.d. from $p(n)$, so for any sampled batch instance $\mathcal{B}_M$, $p(\mathcal{B}_M) = (1/N)^M$. Let $r(\mathcal{B}_M|n)$ denote the probability

---

[2]The same argument holds for the term $\prod_j q(z_j)$ and a similar estimator can be constructed.

[3]This follows from Jensen's inequality $\mathbb{E}_{p(n)}[\log q(z|n)] \leq \log \mathbb{E}_{p(n)}[q(z|n)]$.

of a sampled minibatch where one of the elements is fixed to be $n$ and the rest are sampled i.i.d. from $p(n)$. This gives $r(x_M|n) = (1/N)^{M-1}$.

$$
\begin{aligned}
&\mathbb{E}_{q(z)}\left[\log q(z)\right] \\
=&\mathbb{E}_{q(z,n)}\left[\log \mathbb{E}_{n'\sim p(n)}\left[q(z|n')\right]\right] \\
=&\mathbb{E}_{q(z,n)}\left[\log \mathbb{E}_{p(\mathcal{B}_M)}\left[\frac{1}{M}\sum_{m=1}^{M}q(z|n_m)\right]\right] \\
=&\mathbb{E}_{q(z,n)}\left[\log \mathbb{E}_{r(\mathcal{B}_M|n)}\left[\frac{p(\mathcal{B}_M)}{r(\mathcal{B}_M|n)}\frac{1}{M}\sum_{m=1}^{M}q(z|n_m)\right]\right] \\
=&\mathbb{E}_{q(z,n)}\left[\log \mathbb{E}_{r(\mathcal{B}_M|n)}\left[\frac{1}{NM}\sum_{m=1}^{M}q(z|n_m)\right]\right]
\end{aligned}
\tag{6}
$$

During training, when provided with a minibatch of samples $\{n_1, ..., n_M\}$, we can use the estimator

$$
\mathbb{E}_{q(z)}[\log q(z)] \approx \frac{1}{M}\sum_{i=1}^{M}\left[\log \sum_{j=1}^{M} q(z(n_i)|n_j) - \log(NM)\right]
\tag{7}
$$

where $z(n_i)$ is a sample from $q(z|n_i)$.

### 5.4   FACTORIAL NORMALIZING FLOW

We also performed experiments with a factorial normalizing flow as a flexible prior. Each dimension is a normalizing flow of depth 32, and the parameters are trained to maximize the $\beta$-TCVAE objective. From our preliminary experiments, we found no significant difference between using a factorial Gaussian prior, and so decided not to include this in the main text.

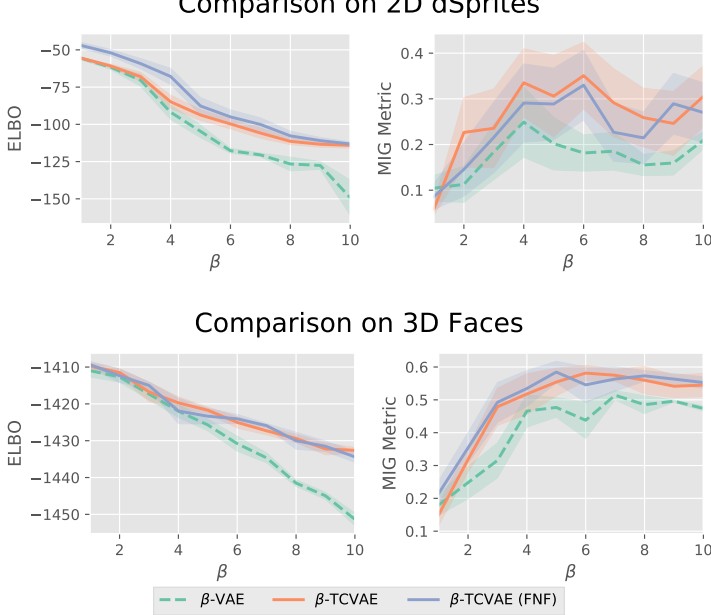

Figure 3: ELBO vs. Disentanglement plots showing the $\beta$-TCVAE with a factorial normalizing flow (FNF).

## 5.5  ESTIMATION OF $I(z_k; v_k)$

With any inference network $q(z|x)$, we can compute the mutual information $I(z; v)$ by assuming the model $p(v)p(x|v)q(z|x)$. Specifically, we compute this for every pair of latent variable $z_j$ and ground truth factor $v_k$.

We make the following assumptions:

- The inference distribution $q(z_j|x)$ can be sampled from and is known for all $j$.
- The generating process $p(n|v_k)$ can be sampled from and is known.
- Simplifying assumption: $p(v_k)$ and $p(n|v_k)$ are quantized (ie. the empirical distributions).

We use the following notation:

- Let $\mathcal{X}_{v_k}$ be the support of $p(n|v_k)$.

Then the mutual information can be estimated as following:

$$
\begin{aligned}
& I(z_j; v_k) \\
=& \mathbb{E}_{q(z_j, v_k)} \left[ \log q(z_j, v_k) - \log q(z_j) - \log p(v_k) \right] \\
=& \mathbb{E}_{q(z_j, v_k)} \left[ \log \sum_{n=1}^{N} q(z_j, v_k, n) - \log q(z_j) - \log p(v_k) \right] \\
=& \mathbb{E}_{p(v_k)p(n'|v_k)q(z_j|n')} \left[ \log \sum_{n=1}^{N} p(v_k)p(n|v_k)q(z_j|n) - \log q(z_j) - \log p(v_k) \right] \\
=& \mathbb{E}_{p(v_k)p(n'|v_k)q(z_j|n')} \left[ \log \sum_{n=1}^{N} \mathbb{1}[n \in \mathcal{X}_{v_k}]p(n|v_k)q(z_j|n) \right] + \mathbb{E}_{q(z_j)} \left[ -\log q(z_j) \right] \\
=& \mathbb{E}_{p(v_k)p(n'|v_k)q(z_j|n')} \left[ \log \sum_{n \in \mathcal{X}_{v_k}} q(z_j|n)p(n|v_k) \right] + H(z_j)
\end{aligned}
\tag{8}
$$

where the expectation is to make sampling explicit.

To reduce variance, we perform stratified sampling over $p(v_k)$, and use $10,000$ samples from $q(n, z_k)$ for each value of $v_k$. To estimate $H(z_j)$ we sample from $p(n)q(z_j|n)$ and perform stratified sampling over $p(n)$. The computation time of our estimatation procedure depends on the dataset size but in general can be done in a few minutes for the datasets in our experiments.

## 5.6  NORMALIZATION

It is known that when $v_k$ is discrete, then

$$
I(z_j; v_k) = \underbrace{H(v_k)}_{} - \underbrace{H(v_k|z_j)}_{\geq 0} \leq H(v_k)
\tag{9}
$$

This bound is tight if the model can make $H(v_k|z_j)$ zero, ie. there exist an invertible function between $z_j$ and $v_k$. On the other hand, if mutual information is not maximal, then we know it is because of a high conditional entropy $H(v_k|z_j)$. This suggests our metric is meaningful as it is measuring how much information $z_j$ retains about $v_k$ regardless of the parameterization of their distributions.

## 5.7  COMPARISON OF DISENTANGLEMENT METRIC PROPERTIES

As summarized in Table 1, our metric detects axis-alignment and is generally applicable and meaningful for any factorized latent distribution, including vectors of multimodal, Categorical, or other structured distributions. This is because the metric is only limited by whether the mutual information can be estimated. Efficient estimation of mutual information is an ongoing research topic Belghazi et al. (2018); Reshef et al. (2011), but we find that the simple estimator (5.5) is sufficient for the datasets we use. We find that MIG can better capture subtle differences in models compared to existing metrics.

| Disentanglement Metric | Axis | Unbiased | General |
|---|---|---|---|
| Higgins et al. (2017) | No | No | No |
| Kim & Mnih (2017) | Yes | No | No |
| MIG (Ours) | Yes | Yes | Yes |

Table 1: In comparison to prior metrics, our proposed MIG detects axis-alignment, is unbiased for all hyperparameter settings, and can be generally applied to any latent distributions provided efficient estimation exists.

## 5.8 COMPARISON OF BEST MODELS

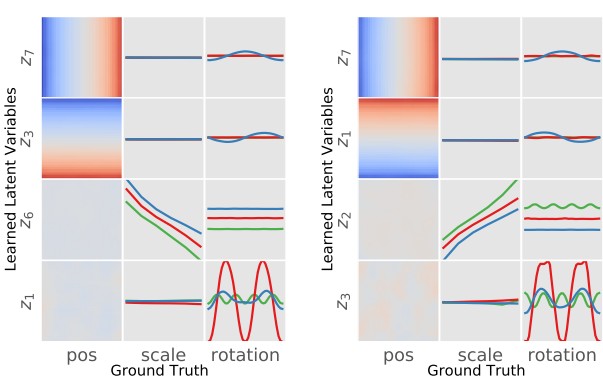

(a) Best $\beta$-TCVAE (MIG: 0.53)     (b) Best $\beta$-VAE (MIG: 0.50)

Figure 4: **MIG is able to capture subtle differences.** Relation between the learned variables and the ground truth factors are plotted for the best $\beta$-TCVAE and $\beta$-VAE on the dSprites dataset according to the MIG metric are shown. Each row corresponds to a ground truth factor and each column to a latent variable. The plots show the relationship between the latent variable mean versus the ground truth factor, with only active latents shown. For position, a color of blue indicates a high value and red indicates a low value. The colored lines indicate object shape with red being oval, green being square, and blue being heart. Interestingly, the latent variables for rotation has 2 peaks for oval and 4 peaks for square, suggesting that the models have produced a more compact code by observing that the object is rendered the same for certain degrees of rotation.

## 5.9 INVARIANCE TO HYPERPARAMETERS

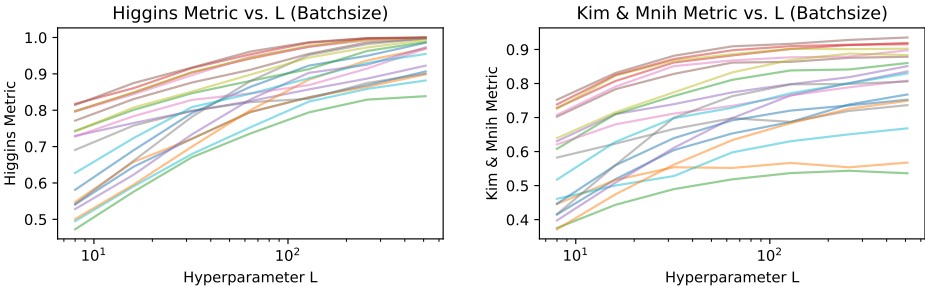

Figure 5: The distribution of each classifier-based metric is shown to be extremely dependent on the hyperparameter $L$. Each colored line is a different VAE trained with the unmodified ELBO objective.

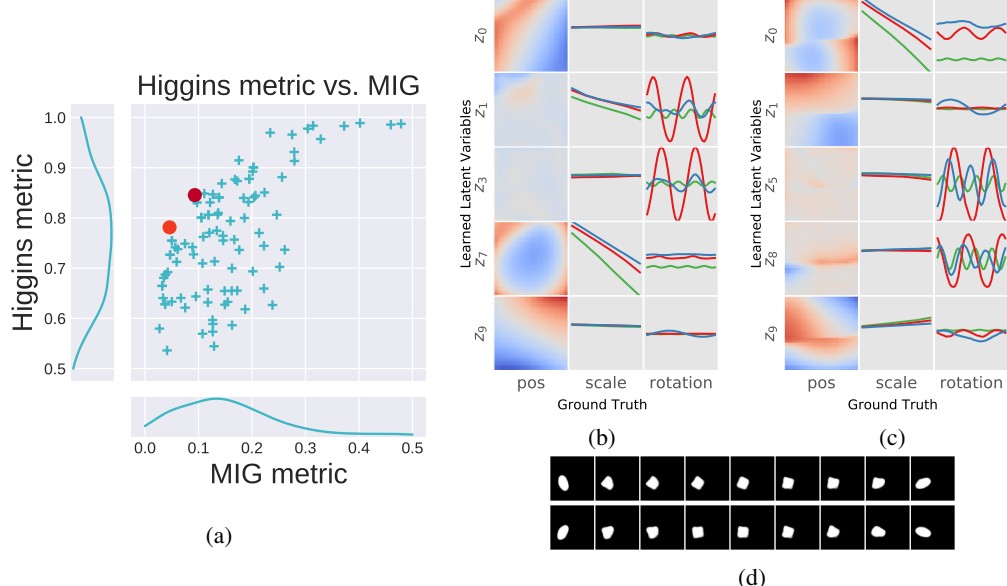

Figure 6: **Entangled representations can have a relatively high Higgins metric while MIG correctly scores it low.** (a) The Higgins metric tends to be overly optimistic compared to the MIG metric. (b, c) Relationships between the ground truth factors and the learned latent variables are shown for the top two controversial models, which are shown as red dots. Each colored line indicates a different shape. (d) Sample traversals for the two latent variables in model (b) that both depend on rotation, which clearly mirror each other.

We believe that a metric should also be invariant to any hyperparameters. For instance, the existence of hyperparameters in the prior metrics means that a different set of hyperparameter values can result in different metric outputs. Additionally, even with a stable classifier that always outputs the same accuracy for a given dataset, the creation of a dataset for classifier-based metrics can still be problematic.

The aggregated inputs used by Higgins et al. (2017) and Kim & Mnih (2017) depend on a batch size $L$ that is difficult to tune and leads to inconsistent metric values. In fact, we empirically find that these metrics are most informative with a small $L$. Figure 5 plots the Higgins et al. (2017) metric against $L$ for 20 fully trained VAEs. As $L$ increases, the aggregated inputs become more quantized. Not only does this increase the accuracy of the metric, but it also *reduces the gap between models*, making it hard to discriminate similarly performing models. The relative ordering of models is also not preserved with different values.

## 5.10 DISAGREEMENTS BETWEEN METRICS

Before using the MIG metric, we first show that it is in some ways superior to the Higgins et al. (2017) metric. To find differences between these two metrics, we train 200 models of $\beta$-VAE with varying $\beta$ and different initializations.

Figure 6a shows each model as a single point based on the two metrics. In general, both metrics agree on the most disentangled models; however, the MIG metric falls off very quickly comparatively. In fact, the Higgins metric tends to output a inflated score due to its inability to detect subtle differences and a lack of axis-alignment.

As an example, we can look at controversial models that are disagreed upon by the two metrics (Figure 6b). The most controversial model is shown in Figure 6a as a red dot. While the MIG metric only ranks this model as better than 26% of the models, the Higgins metric ranks it as better than 75% of the models. By inspecting the relationship between the latent units and ground truth factors, we see that only the scale factor seems to be disentangled (Figure 6b). The position factors are not axis aligned, and there are two latent variables for rotation that appear to mirror each other with

only a very slight difference. The two rows in Figure 6d show traversals corresponding to the two latent variables for rotation. We see clearly that they simply rotate in the opposite direction. Since the Higgins metric does not enforce that only a single latent variable should influence each factor, it mistakenly assigns a higher disentanglement score to this model. We note that many models near the black dot in the figure exhibit similar behavior.

### 5.10.1 MORE CONTROVERSIAL MODELS

Each model is ordered by each metric (MIG or Higgins) such that each model is assigned a unique integer $1-200$. We define the most controversial model as $\max_\alpha R(\alpha; Higgins) - R(\alpha; MIG)$, where a higher rank implies more disentanglement. These are models that the Higgins metric believes to be highly disentangled while MIG believes they are not. Figure 8 shows the top 5 most controversial models.

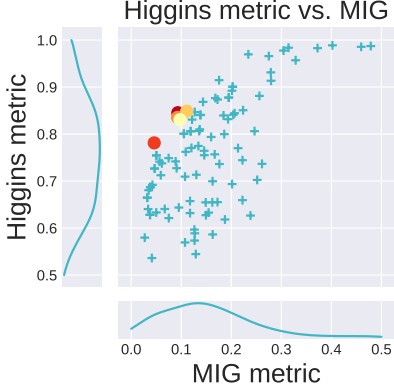

Figure 7: Models as colored dots are those shown in Figure 8.

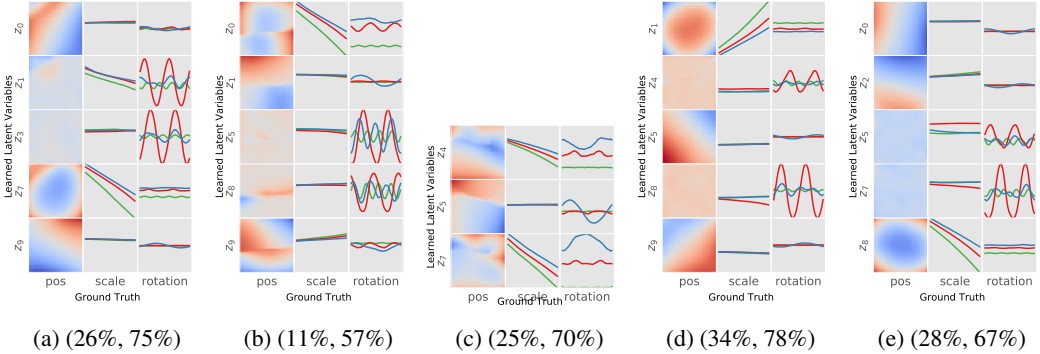

(a) (26%, 75%)  (b) (11%, 57%)  (c) (25%, 70%)  (d) (34%, 78%)  (e) (28%, 67%)

Figure 8: (a-e) The top 5 most controversial models. The brackets indicate the rank of models by MIG and the Higgins metric. For instance, the most controversial model shown in (a) is ranked as better than 75% of model by the Higgins metric, but MIG believes that it is only better than 25% of models.

## 5.11 RANDOM SAMPLES

### 5.11.1 REAL SAMPLES

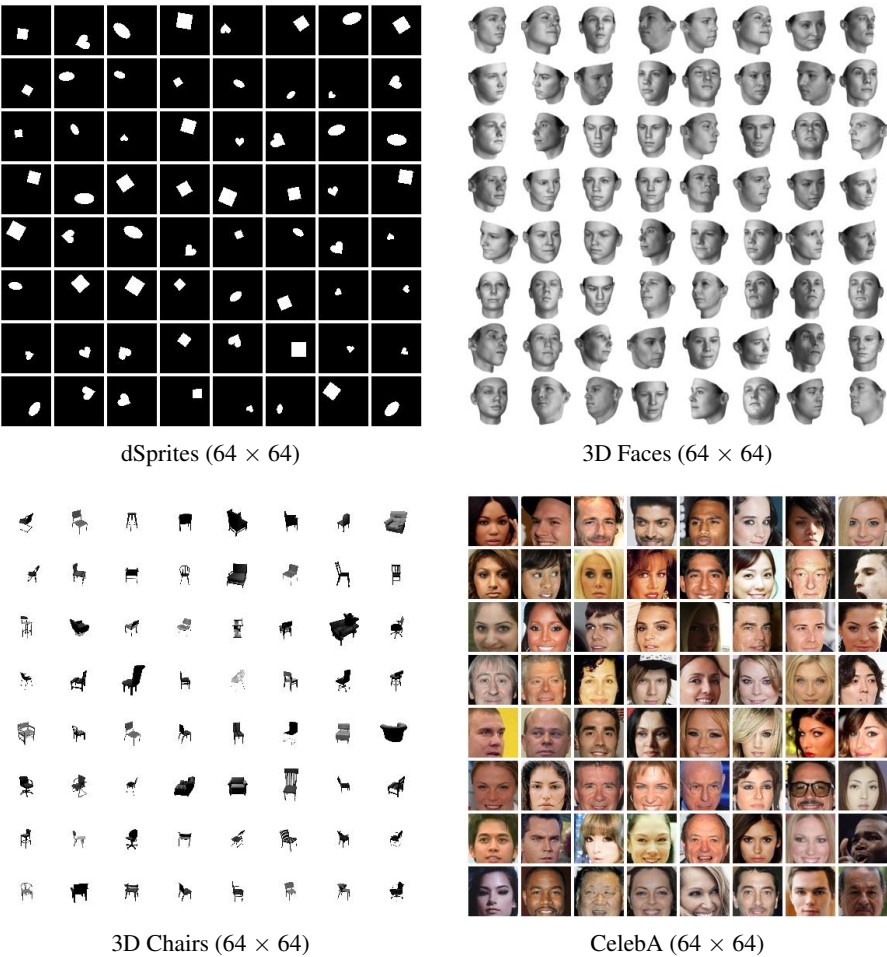

dSprites (64 × 64)    3D Faces (64 × 64)

3D Chairs (64 × 64)    CelebA (64 × 64)

Figure 9: Real samples from the training data set.

### 5.11.2 LATENT TRAVERSALS

$\beta$-TCVAE Model One ($\beta$=15)

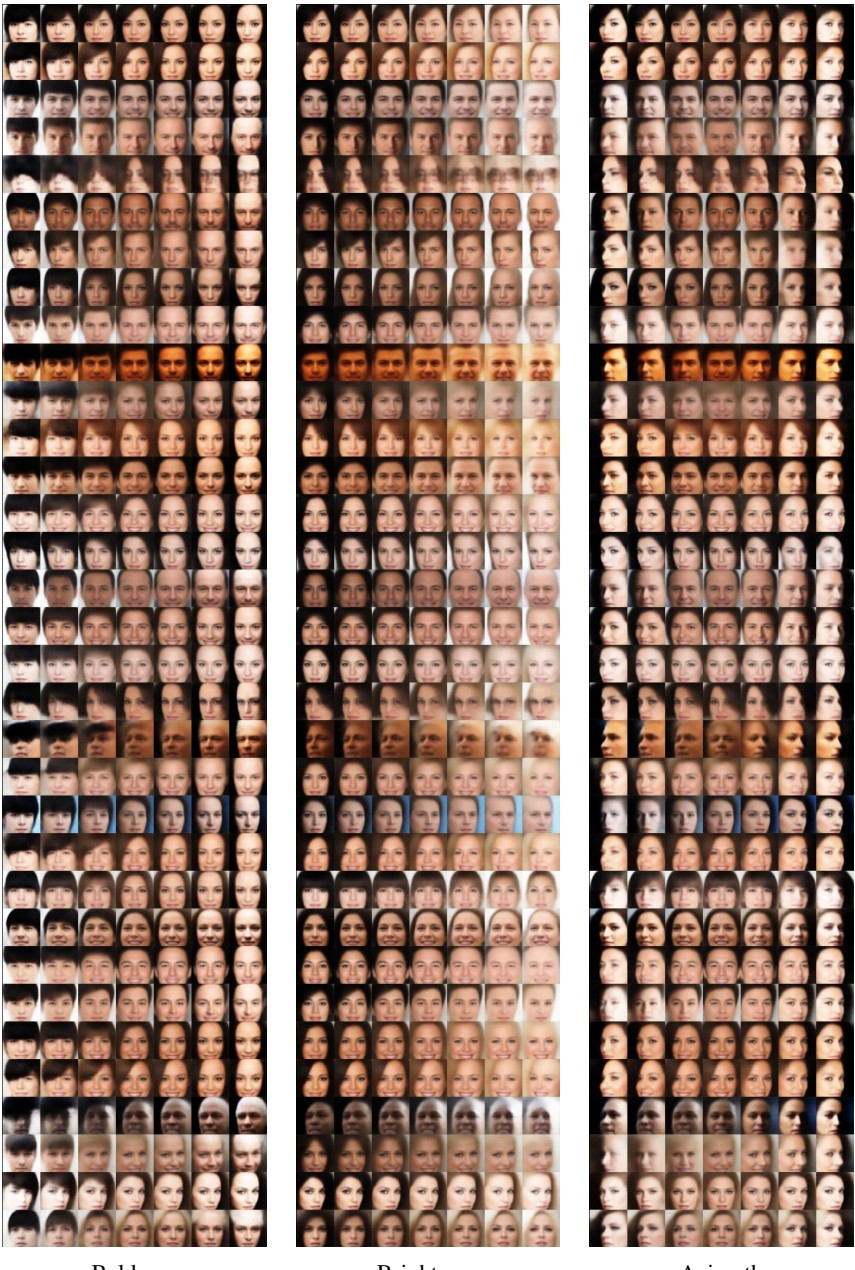

| | | |
|---|---|---|
| Baldness | Brightness | Azimuth |

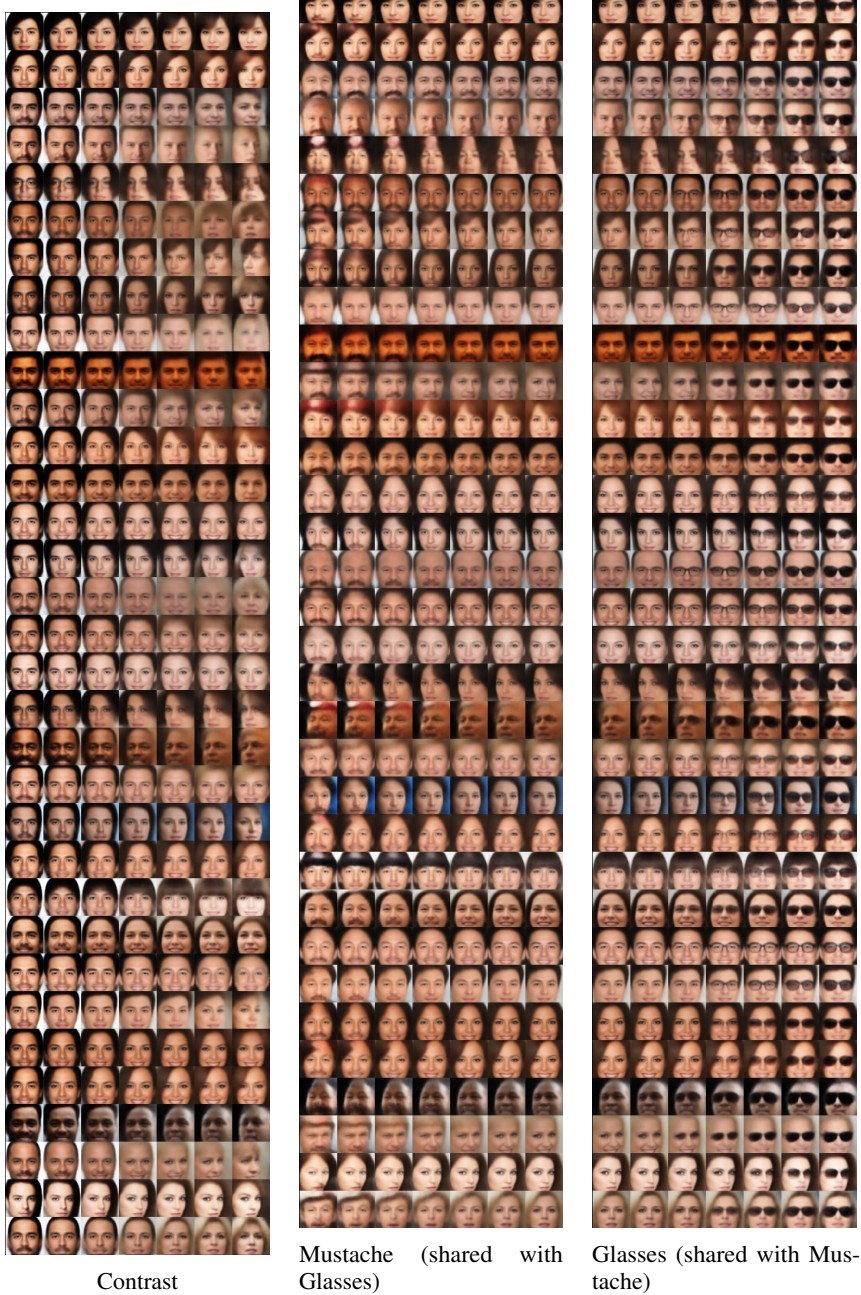

Contrast

Mustache (shared with Glasses)

Glasses (shared with Mustache)

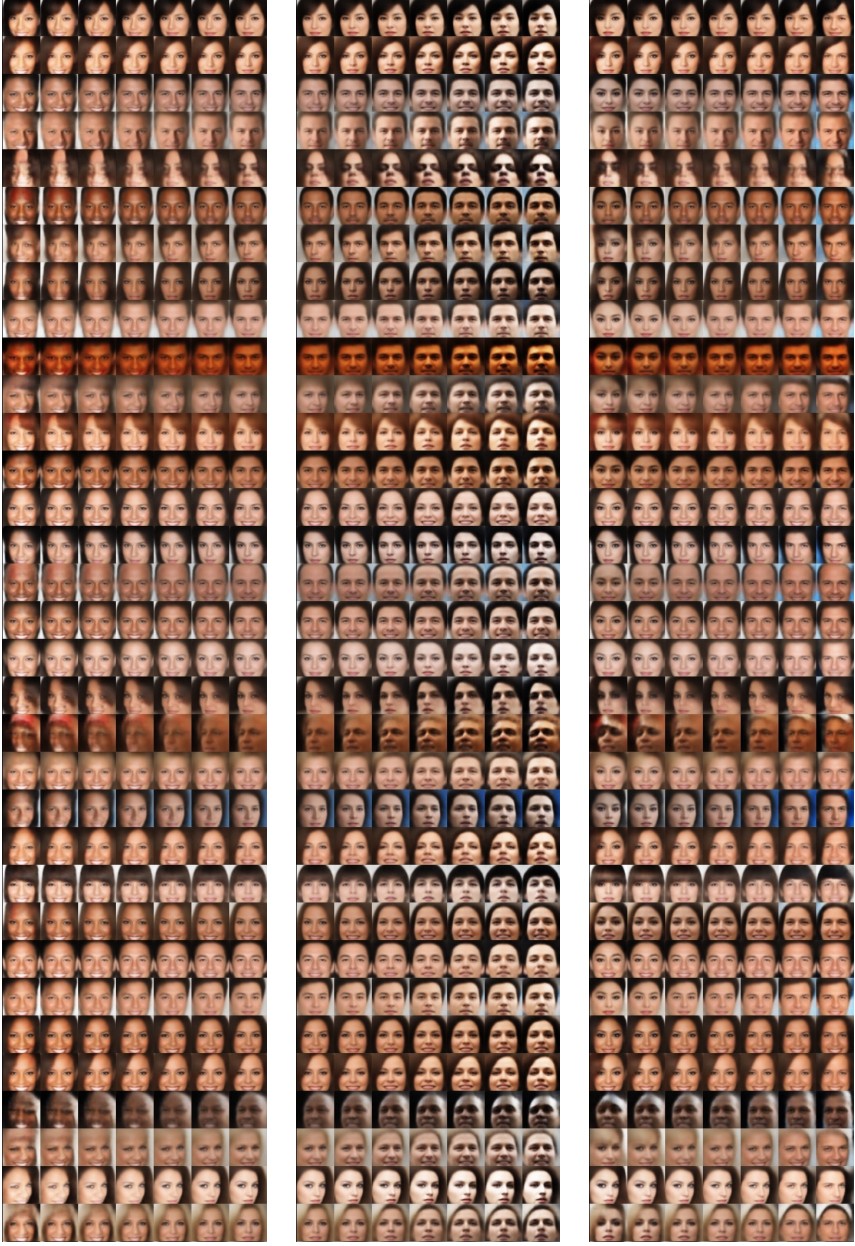

Smile (shared with Shadow)    Shadow (shared with Smile)    Gender

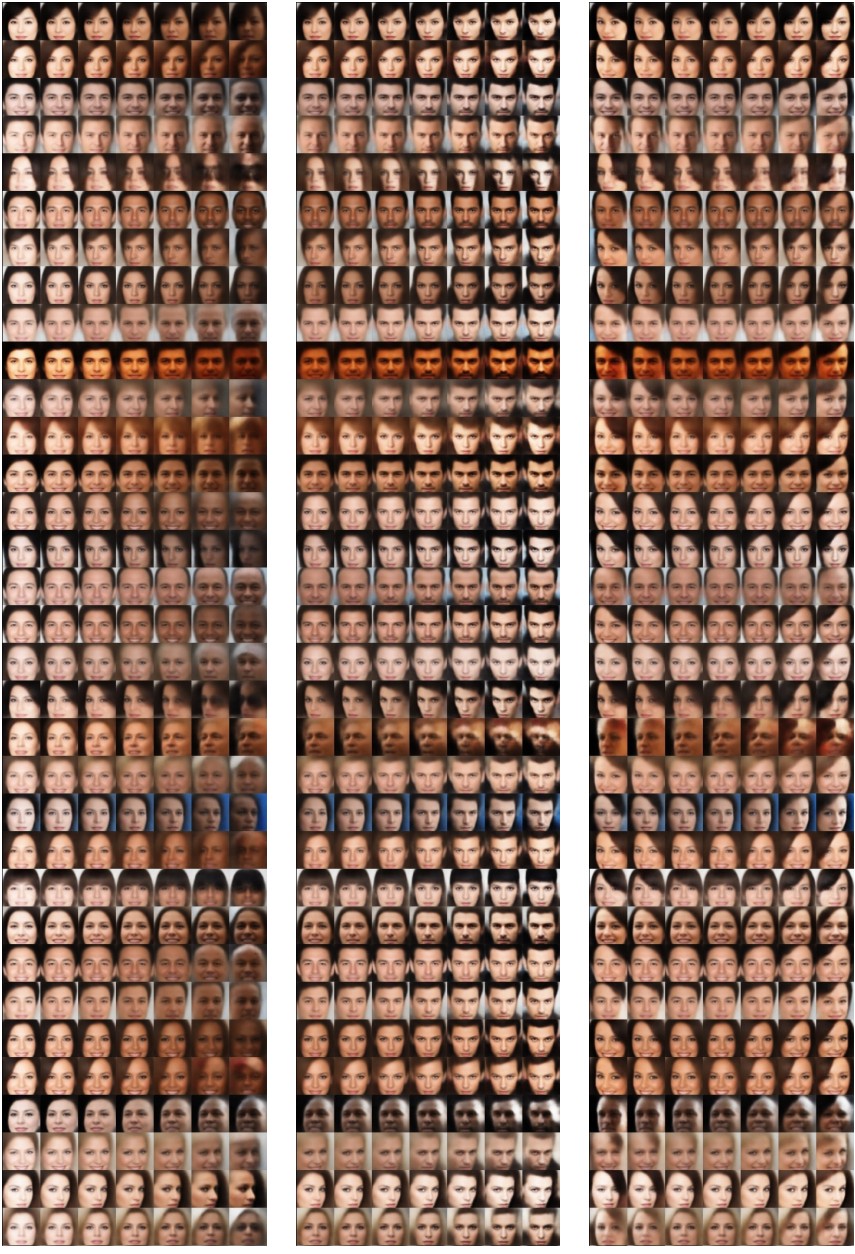

Skin color        Dramatic masculinity        Bangs (side)

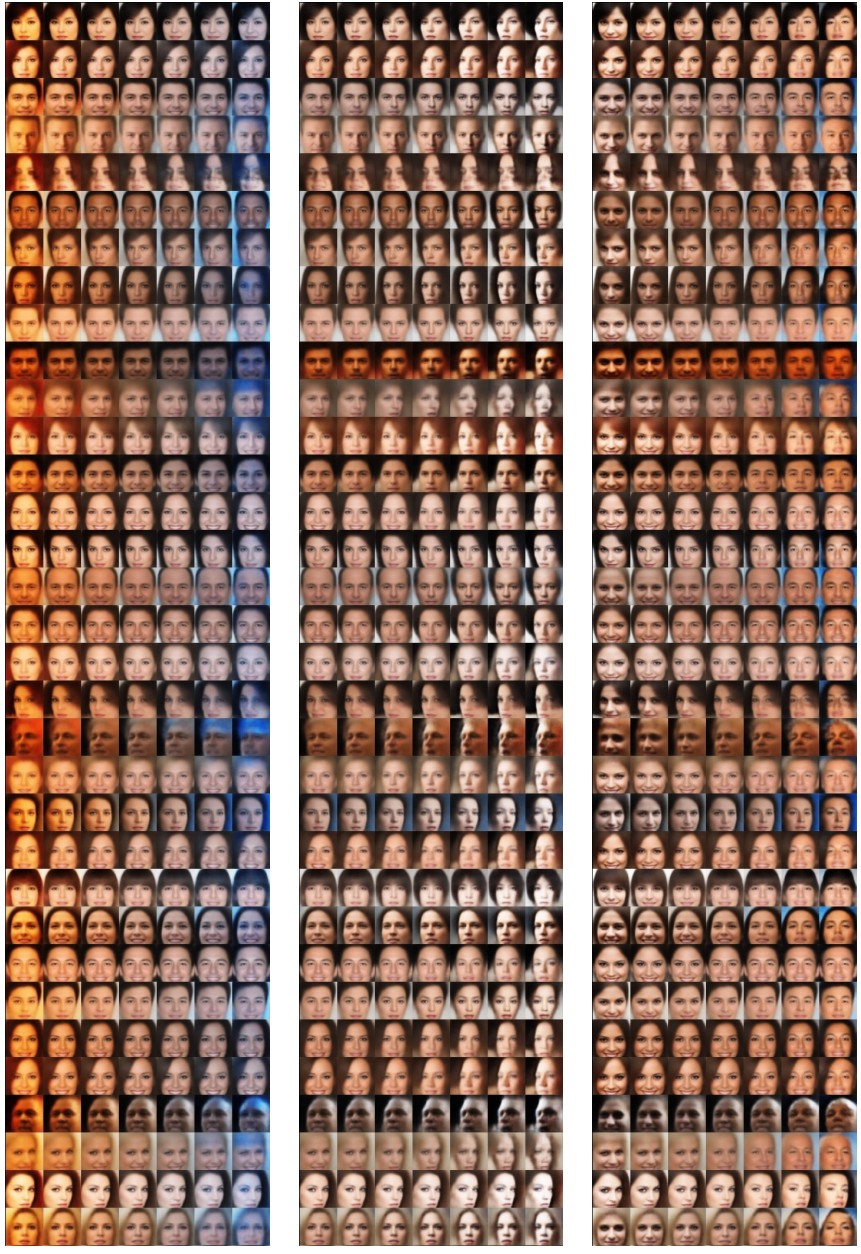

Hue          Face width          Eye shadow

