# OpenReview forum: "Isolating Sources of Disentanglement in Variational Autoencoders"
_ICLR.cc/2018/Workshop — Accept_

### Official Review · AnonReviewer2 · 2018-03-09
**Rewrite and penalize the VAE objective with an explicit total correlation term, proposed a new metric to measure the disentanglement, and improvements over baselines are demonstrated with empirical experiments**

**Rating:** 7
**Confidence:** 3

**Review:**

 [Originality]

The proposed approach is an effective modification of Higgins et al. 2017. Similar ideas based on penalized total correlation have also been presented in Kim and Mnih 2017, but the identification of a total correlation term directly from the ELBO objective seems original.

[Clarity]

This paper is well written and organized.

[Quality]

The experimental results are convincing.

[Significance]

The learning of disentangled representations is an important problem in unsupervised learning and shall have many potential applications.

Pros:
1.	The proposed approach and metric are based on information-theoretic principles
2.	Empirical studies are clearly presented.

Cons: discussion of related work

---

### Official Review · AnonReviewer3 · 2018-03-09
**Interesting paper studying disentanglement.**

**Rating:** 7
**Confidence:** 3

**Review:**

The paper first provides a way to decompose ELBO for variational autoencoders such that correlation between latent can be captured. This explains how BVAE works and helps proposing BTCVAE. Finally authors proposes a new metric MIG for measuring disentanglement. Even in the limited real-estate there is decent quantitative and visual experimental results.

In the appendix, it was interesting to see that there are very different results for different metrics for measuring disentanglement. It would be great to dive into this area in a longer paper.

One thing I didn't like much is the derivation of TCVAE. Intuitively it makes a lot of sense. However ELBO is the method for getting to the latents. The introduced parameters should have some meaning in the original model/auto-encoder.

One thing I would have loved to see is the practical impact of the new  technique. Would this help classify edge cases practically better? For example in any power law distributed images would this help any practical problems such as classification, inverse-problems etc?

---

### Official Review · AnonReviewer1 · 2018-03-12
**Well written paper and important work**

**Rating:** 7
**Confidence:** 4

**Review:**

I read arXiv version of this paper.
It is well written.
The paper proposed the reasonable method based on ELBO decomposition for evaluating the disentanglement quality of representations.
This type of study is important for representation learning because the generation quality of data is typically difficult to evaluate and it is open problem.
The idea of decomposition is interesting and the experiments were well organized.

---

### Public Comment · ~Xuechen_Li1 · 2018-03-07
**Errata**

Due to a mistake, we did not include a comprehensive Related Works section in our draft that gives more emphasis to recent works such as "Disentangling by Factorising" (Kim & Mnih, 2017). We also note that the consistency claim we made in section 5.2 with regards to estimating E_{(q)}[\log q(z)] is not true in general. This, however, does not undermine our empirical analysis.

We will upload a new version when revisions are allowed to address these issues. In addition, our extended version of the paper on arXiv (https://arxiv.org/pdf/1802.04942.pdf) will reflect such changes.

---

### Decision · Program_Chairs · 2018-03-20
**ICLR 2018 Workshop Acceptance Decision**

**Decision:**

Accept

**Comment:**

Congratulations, your paper was accepted to the ICLR workshop.